# Facile Electrochemical Method for the Fabrication of Stable Corrosion-Resistant Superhydrophobic Surfaces on Zr-Based Bulk Metallic Glasses

**DOI:** 10.3390/molecules26061558

**Published:** 2021-03-12

**Authors:** Mengmeng Yu, Ming Zhang, Jing Sun, Feng Liu, Yujia Wang, Guanzhong Ding, Xiubo Xie, Li Liu, Xiangjin Zhao, Haihong Li

**Affiliations:** 1School of Environmental and Material Engineering, Yantai University, No. 30 Qingquan Road, Yantai 264005, China; yumeng51244@163.com (M.Y.); m1391551161@163.com (M.Z.); sunjing469389262@163.com (J.S.); isliufeng@163.com (F.L.); janewangyujia@163.com (Y.W.); dingguanzhong1998@163.com (G.D.); xiuboxie@ytu.edu.cn (X.X.); 2School of Nuclear Equipment and Nuclear Engineering, Yantai University, No. 30 Qingquan Road, Yantai 264005, China; hhli@ytu.edu.cn

**Keywords:** metallic glasses, superhydrophobic surfaces, electrochemical etching, micro–nano composite structures, corrosion resistance

## Abstract

Both surface microstructure and low surface energy modification play a vital role in the preparation of superhydrophobic surfaces. In this study, a safe and simple electrochemical method was developed to fabricate superhydrophobic surfaces of Zr-based metallic glasses with high corrosion resistance. First, micro–nano composite structures were generated on the surface of Zr-based metallic glasses by electrochemical etching in NaCl solution. Next, stearic acid was used to decrease surface energy. The effects of electrochemical etching time on surface morphology and wettability were also investigated through scanning electron microscopy and contact angle measurements. Furthermore, the influence of micro–nano composite structures and roughness on the wettability of Zr-based metallic glasses was analysed on the basis of the Cassie–Baxter model. The water contact angle of the surface was 154.3° ± 2.2°, and the sliding angle was <5°, indicating good superhydrophobicity. Moreover, the potentiodynamic polarisation test and electrochemical impedance spectroscopy suggested excellent corrosion resistance performance, and the inhibition efficiency of the superhydrophobic surface reached 99.6%. Finally, the prepared superhydrophobic surface revealed excellent temperature-resistant and self-cleaning properties.

## 1. Introduction

Since superhydrophobicity was investigated in lotus leaves [1], superhydrophobic surfaces have attracted considerable research attention and have exhibited great application prospects because of their essential properties of self-cleaning [2,3], antifouling [4], antifrosting [5,6], drag reduction [7,8], and oil/water separation [9,10]. Therefore, the fabrication of superhydrophobic surfaces has been widely investigated in various fields such as physics, materials science, and bionics [11,12,13,14].

Generally, a surface with a water contact angle (CA) of >150° is considered to be superhydrophobic [15,16]. It is well known that the superhydrophobicity of solid surfaces is dependent on surface microstructures and surface free energy. Thus, to fabricate artificial superhydrophobic surfaces, it is necessary to increase the surface roughness and lower the surface energy. Normally, micro–nanoscale structures were first developed on a solid surface. Subsequently, the rough surface was modified using low-surface-energy materials. Recently, superhydrophobic surfaces have been extensively fabricated on different substrate materials such as titanium [17,18], magnesium [19,20], aluminium [21], and copper alloys [22,23]. However, the effective transformation of traditional metallic materials with high surface energy to superhydrophobic surfaces remains challenging. Therefore, to broaden the application field of superhydrophobicity, it is necessary to find new substrate materials. Amorphous alloys, also known as metallic glasses, exhibit an atomic arrangement of the short-range order and the long-range disorder, resulting in various excellent properties, such as high strength, hardness, specific strength, and corrosion resistance [24,25,26]. Some methods have been investigated for obtaining superhydrophobic surfaces on different metallic glasses. Ning et al. [27] prepared superhydrophobic surfaces on a Zr_35_Ti_30_Be_26.75_Cu_8.25_ metallic glass by silicon moulding and subsequent chemical etching. Ma et al. [28] conducted thermoplastic forming to fabricate a Pd-based metallic glass surface with micro–nanolayered structures exhibiting superhydrophobicity. Qiao et al. [29] obtained a superhard superhydrophobic iron-based metallic glass coating on a Q235 steel substrate via plasma spraying and surface modification with fluoroalkylsilane (FAS). Although the above-mentioned methods effectively prepared superhydrophobic surfaces, they are still limited by expensive specialised equipment, complicated production processes, and high production costs. Moreover, since FAS is harmful to the human body, its use is not conducive for biological applications.

In addition to the glass-forming ability of a metallic glass, the toxicity of the contained elements, such as Ni and Be, must be considered in order to expand the practical applications. Moreover, numerous studies have suggested that Zr-Cu-Al metallic glasses exhibit high biocompatibility [30,31,32]. Therefore, the Zr_46_Cu_46_Al_8_ metallic glass was employed to fabricate superhydrophobic surfaces. In this work, a safe, simple, and inexpensive method was proposed to prepare superhydrophobic surfaces. First, electrochemical etching was performed to generate micro–nano composite structures on the surface of Zr-based metallic glasses. Next, the obtained rough surfaces were further modified via immersion in a stearic acid solution. Compared with other methods, the electrochemical corrosion selected in this work displays the advantages of being easily performed, relatively fast, very reproducible, low-cost, and suitable for the large-scale manufacturing of superhydrophobic surfaces. The effects of electrochemical etching time on the surface morphology and wettability were investigated. The corrosion resistance of the as-prepared superhydrophobic surfaces was investigated via Tafel polarisation and electrochemical impedance spectroscopy (EIS). The chemical stability of the superhydrophobic surface was studied through high temperature and different pH solutions. Furthermore, the self-cleaning properties of the prepared superhydrophobic surfaces were examined. Considering that Zr_46_Cu_46_Al_8_ metallic glasses have high biocompatibility, and the stearic acid surface modification solution does not contain fluorine that is harmful to the human body, the proposed method is advantageous for fabricating superhydrophobic surfaces and can have promising applications in biomedicine.

## 2. Experimental

### 2.1. Preparation of Zr-Based Metallic Glasses

The bulk metallic glasses specimens (10 mm × 1.5 mm × 70 mm) were manufactured from high-purity metals of 46 at.% Zr, 46 at.% Cu, and 8 at.% Al by vacuum arc-melting and copper-mould casting in an argon atmosphere. Subsequently, the specimens were cut into 10 mm × 1.5 mm × 10 mm samples by a diamond cutter and polished on 800 mesh, 1000 mesh, and 2000 mesh sandpapers. Finally, the samples were ultrasonically washed with ethyl alcohol for 5 min, followed by drying.

### 2.2. Fabrication of Superhydrophobic Surfaces

The as-prepared samples were used as the anode, while a platinum plate of the same size was used as the cathode, which was located 15 mm from the anode. Electrochemical etching was carried out in 3.5 wt.% NaCl solution at a current density of 0.2 A cm^−2^ for 2–22 min. This process was carried out at 35 °C under ultrasonic conditions. Next, the etched samples were ultrasonically rinsed with deionised water, ethyl alcohol, and acetone, followed by air-drying. Finally, they were modified by immersion in 8 mmol/L stearic acid at room temperature for 20 h, followed by heating at 120 °C for 2 h.

### 2.3. Sample Characterisation

The Zr-based metallic glasses were structurally characterised by X-ray diffraction (XRD) using a diffractometer (7000X, Shimadzu, Japan) with Cu-Kα radiation. The microstructures of the as-prepared superhydrophobic samples were observed by scanning electron microscopy (SEM; JSM-7610F, JEOL, Tokyo-to, Japan). The chemical compositions were determined by energy-dispersive spectroscopy (EDS; JSM-7610F, JEOL, Japan). Surface roughness was measured by a surface roughness meter (Ra, RTP-130, Shanghai, China). CA was measured by the sessile drop method at room temperature by using a static CA meter (CA100C, Innuo, Shanghai, China). Water droplets (5 μL) were dropped on five randomly selected positions on the sample. The sliding angle was closely related to surface adhesion. A water droplet (5 μL) was placed on the surface fixed to a tiltable platform primitively at the horizontal position. As the platform gradually inclined, the critical inclination angle was considered as the sliding angle.

The potentiodynamic polarisation test and EIS were performed in 3.5 wt.% NaCl solution at room temperature by using a three-electrode electrochemical workstation (CHI660E, Chenhua, China). The saturated calomel electrode was used as a reference electrode, while the platinum plate was used as a counter electrode. Before polarisation and EIS measurements, the open-circuit voltage was measured in 3.5 wt.% NaCl solution. The polarisation curves of the samples were obtained at a scan rate of 2 mV s^−1^. For EIS, the frequency range was 0.01–100,000 Hz, and the amplitude of the sinusoidal signal was 5 mV.

In order to evaluate the stability of the sample surface, HCl and NaOH were used to prepare solutions with different pH, and their droplets were dropped onto the surface of the superhydrophobic sample. After standing for 5 min, the contact angle between the droplets and the superhydrophobic surface was measured. To further evaluate the durability of the superhydrophobic surface, the samples were placed in an oven at different temperatures for 2 h, and the change in CA was measured.

To prove the self-cleaning ability of the superhydrophobic surface, quartz sand was evenly covered on the sample surface, and water droplets dripped from the top of the inclined surface. In addition, the self-cleaning performance of the superhydrophobic surface was further confirmed by the pulling experiment. This time, sand was placed on the sample surface, and 5 μL of water droplets was suspended on the probe, which was then moved down to wrap the particles in the water droplets and lifted up afterwards.

## 3. Results and Discussion

### 3.1. Wettability Analysis and Surface Morphology

Experimental data show that the surface of the Zr-based metallic glasses is hydrophilic before electrochemical etching. The superhydrophobicity of the samples was attributable to etching time and surface modification. The behaviour of the sample surfaces at different electrochemical etching times can be more significant when investigating superhydrophobic properties. Table 1 shows the roughness, CA, and sliding angle of the samples for different etching times. When the etching time is within 15 min, with the increase of time, CA, and roughness of the samples gradually increase. However, when the etching time exceeds 15 min, the sample excessively corrodes and CA decreases. Thus, according to experimental data, 15 min can be considered as the optimal electrochemical etching time for fabricating superhydrophobic surfaces.

A complex contact idea proposed by Cassie and Baxter [33] assumes that when a water droplet is in contact with a solid surface, it forms a composite contact, combining solid–liquid contact with gas–liquid contact. The expression is as follows: $${\cos\theta }_{c}{\;=\;f}_{s}\;\left(\cos\theta \right.+\;\left.1\right)-1$$
where f_s_ is the area fraction occupied by the solid–liquid contact surface. It can be analysed from the formula that the smaller the f_s_, the larger the contact angle. Accordingly, under the combined effect arising from the presence of micro–nano composite structures and the surface tension of water droplets, cavitation occurs on the rough surface. The trapped air leads to a decrease in f_s_, which improves the wettability and increases the contact angle. Figure 1 shows SEM images of the samples after surface modification at different etching times, which visually illustrate the microscopic morphology of the sample surface [34]. Figure 1a demonstrates that the entire sample surface was characterised by its network structures, and the roughness of the surface was 0.692 μm. However, the single grid actually served as the etch pit, and the micrometre-scale structures were observed in it at high magnification (Figure 1b). At the etching time of 2 min, the microstructures did not completely cover the surface of the sample, and the number of air pockets generated was limited, which resulted in the sample surface being infiltrated by the droplets. Moreover, the surface with lower roughness did not facilitate sufficient adhesion with stearic acid. At the moment, CA was only 138.6°; thus, the surface could not achieve superhydrophobicity. When the electrochemical etching time was 7 min, the surface morphology was characterised by microstructures in the shape of a coral reef (Figure 1c). Furthermore, micro–nanoscale mastoids were found at ×20,000 magnification (Figure 1d). These coral-reef-shaped micro–nano composite structures not only increased the roughness of the sample surface but also prevented surface wetting because of the occurrence of cavitation. Thus, after surface modification, the superhydrophobic surface was prepared with a water CA of 153.4° and a roughness of 2.303 μm. At the etching time of 15 min, although the surface morphology reflected the presence of coral-reef-shaped micro–nano composite structures (Figure 1e), the number of nanoscale mastoids increased significantly (Figure 1f). At this time, the roughness of the sample surface was 4.566 μm, which is almost twice as large as that at 7 min. According to the Cassie–Baxter expression, when the surface roughness increased, more air was trapped on the sample surface, so the f_s_ decreased and the contact angle increased. In addition, the dense coral reef-like structure provided a uniform air cushion on the sample surface, resulting in a smaller solid–liquid contact area on the sample surface. Hence the contact angle reached 154.3° after modification. After electrochemical etching for 22 min, due to the combined effect of electrochemical etching and ultrasonic vibration, cracks appeared on the surface (Figure 1g). At ×10,000 magnification, the micro–nano composite structures disappeared (Figure 1h). Since the contact area of the droplet with the surface increased, hydrophobicity decreased remarkably. Thus, CA was only 151.2°.

### 3.2. Chemical Composition and Surface Chemical Reaction Process

Electrochemical etching not only increased surface roughness but also changed surface chemical composition. An oxidation reaction occurred at the anode of the sample as it was electrochemically etched. Al, Zr, and Cu successively lost electrons and dissolved, forming pits on the surface of the sample, and then Cu quickly deposited back into the pits [35]. Finally, together with the oxidation products, micro–nano composite structures were formed on the surface of the sample. The chemical composition of the sample surface before and after the reaction was characterised by XRD. Due to the long-range disorder and short-range order of amorphous materials, their XRD patterns significantly differ from crystalline materials [36]. Figure 2a represents the XRD pattern of the sample surface before the reaction, showing that the sample exhibited only one scattering peak. In contrast, Figure 2b shows that the sample surface has characteristic peaks at 43.5°, 50.7°, and 74.7°, corresponding to the crystal planes of Cu (111), Cu (200), and Cu (220), respectively. The above data are consistent with the crystallisation data of Cu (JCPDS card: 01-1242), indicating that there is copper on the surface of the sample after electrochemical etching. The reaction process for the Zr_46_Cu_46_Al_8_ metallic glass in the entire electrochemical system was presumed as follows:

Al_(s)_ + 3Cl^−^
_(aq)_ = AlCl_3 (aq)_ + 3e^−^

Zr_(s)_ + 4Cl^−^
_(aq)_ = ZrCl_4 (aq)_ + 4e^−^

Cu_(s)_ + Cl^−^
_(aq)_ = CuCl _(s)_ + e^-^

2CuCl _(s)_ + H_2_O _(l)_ = Cu_2_O _(s)_ + 2Cl^−^
_(aq)_ + 2H^+^
_(aq)_

2Cl^−^ + 2e^−^ = Cl_2_ ↑

Cu_2_O _(s)_ + 2H_2_O _(l)_ + O_2_ = 2CuO _(s)_ + 2H_2_O_2 (l)_

Cu_2_O _(s)_ + 2H_2_O_2 (l)_ = 2CuO _(s)_ + 2H_2_O _(l)_

2H^+^ + 2e^−^ = H_2_ ↑

The above reaction process implies that the preparation method is safe, simple, and environmentally friendly. The final products of redox were H_2_ and a mixture of Cu, CuO and Cu_2_O. Further, Cl_2_ was dissolved in the NaCl solution, and the final electrolyte solution was weakly alkaline. Therefore, the method does relatively little harm to the environment.

Figure 3 represents the EDS spectrum of the sample whose surface was modified by stearic acid. It reveals that the surface is composed of Zr, Cu, C, and O, a small quantity of Al, and traces of Cl. As expected, Zr, Cu, and Al are the elements contained in the sample; Cl may also be the reaction product or the residue of NaCl solution treatment; O is from the oxide produced during electrochemical etching and the element in the stearic-acid-modified self-assembled film; and C arises from the component of the membrane self-assembled on the surface owing to stearic acid treatment. After chemical modification, self-assembled monolayers (SAMs) of stearic acid are formed on the surface of the sample. The hydrophilic polar head-group (-COOH) of stearic acid is tightly bonded to the substrate through chemisorption, leading to the hydrophobic end-group (-CH_3_) staying away from the surface and decreasing surface energy. The reaction processes may be as follows [37,38]:

Al^3+^ + 3CH_3_(CH_2_)_16_COOH = Al[CH_3_(CH_2_)_16_COO] + 3H^+^

Zr^4+^ + 4CH_3_(CH_2_)_16_COOH = Zr[CH_3_(CH_2_)_16_COO] + 4H^+^

Cu^2+^ + 2CH_3_(CH_2_)_16_COOH = Cu[CH_3_(CH_2_)_16_COO] + 2H^+^

### 3.3. Corrosion Resistance

The potentiodynamic polarisation test is one of the main methods for the characterisation of the corrosion resistance of a sample. In the polarisation curve, the corrosion resistance of the sample primarily depends on the corrosion potential (E_corr_) and the corrosion current density (i_corr_). The higher the corrosion potential and the lower the corrosion current density, the better the corrosion resistance of the sample [39]. Figure 4a shows the polarisation curves of samples with different etching times in 3.5 wt.% NaCl solution. As expected, the corrosion resistance of the sample with an etching time of 15 min was higher than that of the original sample and samples with other etching times. Furthermore, the samples with other etching times also exhibited better corrosion resistance than the original. This is related to the superhydrophobicity of the sample surface. Table 2 shows the corrosion potential and the corrosion current density of the sample obtained by the Tafel curve extrapolation method [40,41]. It can be seen from Table 2 that the corrosion potential of the sample with an etching time of 22 min was not significantly different from that of the original sample; however, the corrosion current density was reduced by one order of magnitude. In addition, the corrosion potential of the superhydrophobic samples subjected to 7 and 15 min of etching in 3.5 wt.% NaCl solution was increased by 100 and 200 mV, respectively, and the corrosion current density was reduced by one order and three orders of magnitude, respectively. Similarly, the corrosion resistance of samples can be explained by inhibition efficiency (η) [42].
$${\eta \;(\%)\;=\;(1\;-\;I}_{0}/I_{\mathrm{corr}})\text{ × }100$$
where I_0_ and I_corr,_ are the corrosion current densities for the modified samples and original sample, respectively. The inhibition efficiency of the 15 min sample was 99.6%, and that of other samples with different etching times was less than 90%. Consequently, the sample with the electrochemical etching time of 15 min exhibited the best corrosion resistance. The micro–nano composite structures of the superhydrophobic surface intercepted the air under the synergistic effect of surface tension and capillary force to form air pockets. Therefore, the surfaces of the sample and the solution were separated by air, and the corrosive medium could not contact the sample. Furthermore, given that the alkyl chain was characterised by nonconductivity and hydrophobicity, an energy barrier for ion conduction and electron tunnelling was established by dense SAMs, which prevented electrons from passing through the electrode interface. Therefore, the contact between corrosive ions and the sample surface was effectively blocked, and the corrosion process was suppressed.

EIS is also a highly effective method for testing the corrosion resistance of samples and can complement the polarisation curve. The potentiodynamic polarisation test results demonstrated that the samples with a corrosion time of 15 min have the best corrosion resistance; therefore, EIS was used to discuss its anticorrosion performance further. Figure 4b presents Nyquist plots of the original sample and the superhydrophobic sample measured by frequency scanning in 3.5 wt.% NaCl solution. The data fitted by ZView2 software are shown in Table 3, and the equivalent circuit used is provided in Figure 4d. The Nyquist curve of the sample is a capacitive arc in the high-frequency region, indicating that the corrosion behaviour of the sample is controlled by charge transfer. The larger the radius of the capacitive reactance arc, the greater the resistance of the charge when it is transferred, and the better the corrosion resistance of the sample [43]. Figure 4b shows that the capacitive reactance arc of the superhydrophobic sample is much larger than that of the original sample. Moreover, the impedance value of the original sample obtained by the simulation is 16,084 Ω⋅cm^2^, and that of the 15 min superhydrophobic sample is 85,518 Ω⋅cm^2^; accordingly, the superhydrophobic sample exhibits better corrosion resistance. As the synergy effect between the barrier properties of air pockets and SAMs in the superhydrophobic surface, charges and corrosive media are quarantined, thereby improving the corrosion resistance of the superhydrophobic sample.

Figure 4c represents Bode plots of the impedance spectrum of the original sample and the superhydrophobic sample in the electrolyte solution. The Bode diagram illustrates that the impedance value of the superhydrophobic sample is higher than that of the original sample, the maximum phase angle of the superhydrophobic sample is larger than that of the original sample and its corresponding frequency range is wider than that of the original sample. The above two points indicate that the corrosion resistance of the superhydrophobic sample outperforms that of the original sample. The comparison of the Tafel curve and the electrical impedance spectrum of the original sample with the superhydrophobic sample demonstrates that the superhydrophobic sample has better corrosion resistance.

### 3.4. Chemical Stability

Figure 5a shows the contact angles of different pH solutions with the sample surface. Their contact angles are not much different, and they all maintain a superhydrophobic state. This is because the dual role of SAMs and the surface air layer prevents the solution from wetting the superhydrophobic surface. Figure 5b reflects the changing trend of CAs of the sample after heating at different temperatures for 2 h. The superhydrophobic surface has thermal stability given that it remains superhydrophobic after being placed in a 210 °C oven for 2 h.

### 3.5. Self-Cleaning Properties

The superhydrophobic surface is proven to be self-cleaning if it allows droplets to take away surface contaminants while rolling down. Figure 6a shows the self-cleaning process of the sample on an inclined surface. The surface of the sample was covered with a uniform layer of quartz sand, and water drops dripped from the top of the inclined surface, and the quartz sand was taken away in the process of rolling down. Furthermore, Figure 6b also displays the process of self-cleaning the surface particles on the horizontal plane. The quartz sand was placed on the surface of the sample, then 5 μL of water drop was suspended on the probe, moving the probe downward so that the particles were wrapped in the water droplets. The probe was lifted, and the particles were also carried away from the sample surface. Water droplets can remove contaminants without wetting the sample surface, which can be attributed to the low adhesion of the superhydrophobic surface. Because the micro–nano composite structures trap the air, the contact area between the droplet and the surface is significantly reduced, and there is also the effect of capillary force. Therefore, the self-cleaning ability can prevent the sample from being attached to liquid or other contaminants and has a certain protective effect.

## 4. Conclusions

A safe, environmentally friendly, and inexpensive method was proposed to prepare superhydrophobic surfaces on the Zr_46_Cu_46_Al_8_ metallic glass. By this method, micro–nano composite structures were generated on the surface of the samples by electrochemical etching in 3.5 wt.% NaCl solution. The surface energy of samples was subsequently reduced by immersion in 8 mmol/L stearic acid. The potentiodynamic polarisation test and EIS confirmed that the superhydrophobic sample exhibits high corrosion resistance with an inhibition efficiency of 99.6%. In addition, the superhydrophobic sample reveals excellent chemical stability and self-cleaning properties. Meanwhile, the prepared superhydrophobic sample contains no harmful elements. Therefore, the proposed method has wide-scale applications in various fields, particularly biomedicine.

## Figures and Tables

**Figure 1 molecules-26-01558-f001:**
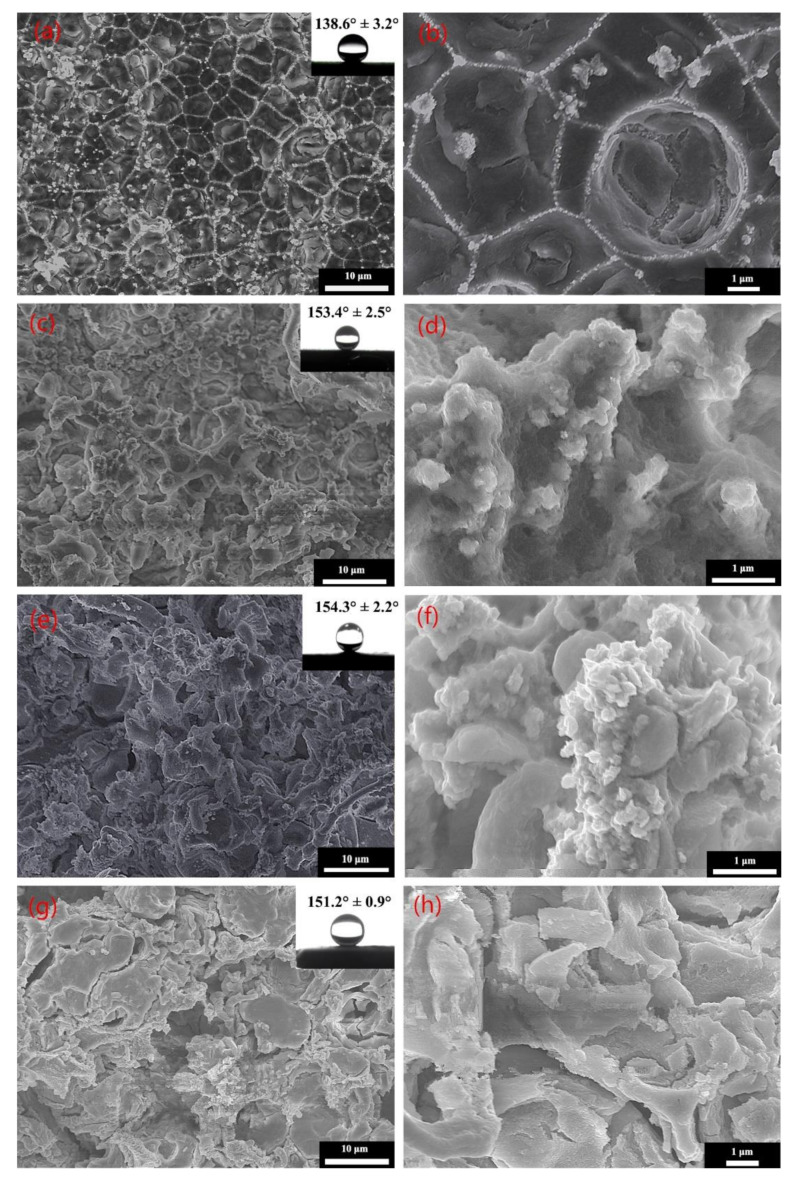
SEM images of the sample surfaces at different processing times: (**a**,**b**) 2 min, (**c**,**d**) 7 min, (**e**,**f**) 15 min, and (**g**,**h**) 22 min.

**Figure 2 molecules-26-01558-f002:**
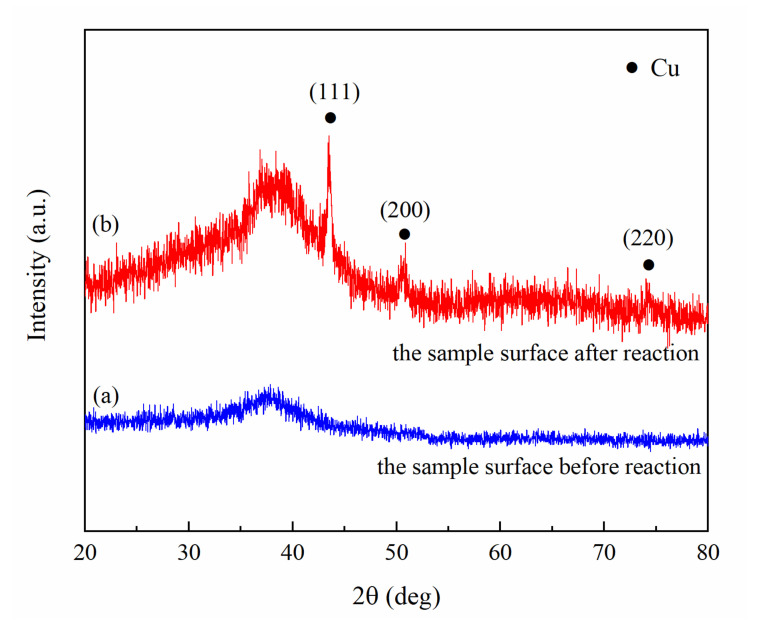
XRD pattern of the sample surface before (**a**) and after (**b**) reaction.

**Figure 3 molecules-26-01558-f003:**
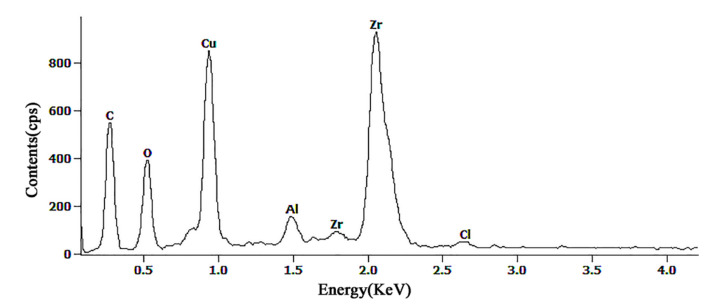
Surface energy spectrum of the surface after modification.

**Figure 4 molecules-26-01558-f004:**
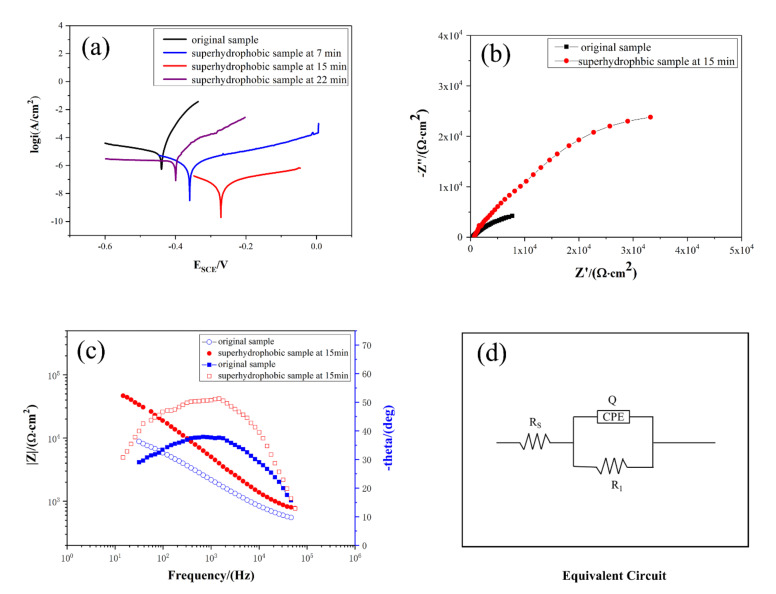
(**a**) Polarisation curves of samples in different corrosion times in 3.5 w.t.% NaCl solution; (**b**) Nyquist plots of the original sample and the superhydrophobic sample in a 3.5 wt.% NaCl solution; (**c**) bode plots of the impedance spectrum of the original sample and the superhydrophobic sample in a 3.5 wt.% NaCl solution; (**d**) equivalent circuit diagram.

**Figure 5 molecules-26-01558-f005:**
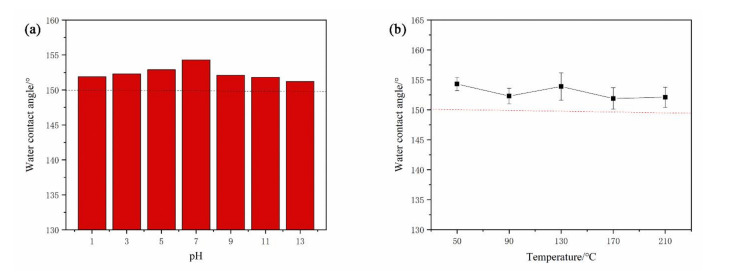
(**a**) CAs of different pH solutions; (**b**) CAs of superhydrophobic samples at different temperatures.

**Figure 6 molecules-26-01558-f006:**
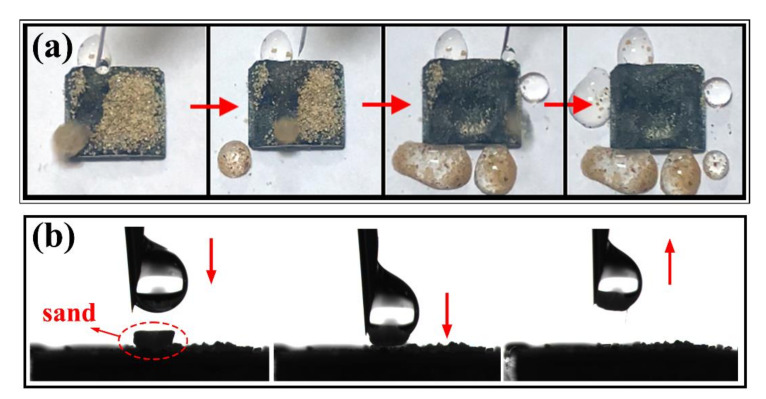
(**a**) Self-cleaning process of the sample on an inclined surface; (**b**) the process of self-cleaning the surface particles on the horizontal plane.

**Table 1 molecules-26-01558-t001:** Contact angle (CA), sliding angle and roughness of Zr-based metallic glasses at different electrochemical etching times.

Etching Time (min)	Roughness (μm)	CAs after Electrochemical Etching	CAs after Surface Modification	Sliding Angle
0	0.452	75.3° ± 1.2°	106.7° ± 1.5°	> 10°
2	0.692	108.6° ± 4.2°	138.6° ± 3.2°	> 10°
7	2.303	114.3° ± 2.9°	153.4°± 2.5°	< 5°
12	3.550	119.4° ± 4.3°	153.2° ± 3.0°	< 5°
15	4.566	116.3° ± 3.9°	154.3° ± 2.2°	< 5°
17	4.916	123.1° ± 5.8°	152.4° ± 2.4°	< 5°
22	5.322	116.3° ± 3.3°	151.2° ± 0.9°	< 5°

**Table 2 molecules-26-01558-t002:** Corrosion potential and corrosion current density of samples with different electrochemical etching times.

Electrochemical Etching Time (min)	E_corr_ (V)	I_corr_ (A cm^−2^)	η (%)
0	−0.472	1.080 × 10^−5^	-
7	−0.370	1.376 × 10^−6^	87.3
15	−0.280	3.920 × 10^−8^	99.6
22	−0.415	2.160 × 10^−6^	80.0

**Table 3 molecules-26-01558-t003:** Impedance parameters obtained by fitting the impedance spectrum.

Etching Time (min)	R_s_ (Ω⋅cm^2^)	Q_CPE_ (Ω^−1^s^−n^ cm^−2^)	n	R_1_ (Ω⋅cm^2^)
0	390.4	3.48′ 10^−6^	0.567	16,084
15	593.2	6.83 × 10^−7^	0.659	85,518

## Data Availability

The data presented in this study are available in this article.

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
