# Peer review of "Facile Electrochemical Method for the Fabrication of Stable Corrosion-Resistant Superhydrophobic Surfaces on Zr-Based Bulk Metallic Glasses"

_molecules, 2021, doi:10.3390/molecules26061558_

Round 1

Reviewer 1 Report

Journal: Molecules

Ms. Ref. No.: molecules-1116757

Title: Facile electrochemical method for fabrication of stable corrosion-resistant superhydrophobic surface on Zr-based bulk metallic glasses

This is an interesting work that reports a facile electrochemical method for fabrication of stable corrosion-resistant superhydrophobic surface on Zr-based bulk metallic glasses. The corrosion-resistant and superhydrophobic properties were dramatically improved by the electrochemical etching method. The experiments were rationally designed. This work provides a new strategy for improving corrosive and superhydrophobic properties of bulk metallic glasses. It is acceptable in this journal after a minor revision.

  1. The surface morphology evolution at different etching time is characterized by SEM. However, the structural evolution and surface elemental composition are missed out. The XRD pattern of the etched sample and elemental composition of original sample should be provided, such as in [Adv. Mater. 32 (2020) 2000385; Prog. Mater. Sci. 105 (2019) 100576].

  1. The quality of the Figures should be further improved, such as the name of X axis in Figure 4b is missing. In addition, there are several typos and mistakes in the present manuscript. Please carefully correct the whole errors. Please do not use an image as the Equation in the manuscript.

Author Response

  1. The surface morphology evolution at different etching time is characterized by SEM. However, the structural evolution and surface elemental composition are missed out. The XRD pattern of the etched sample and elemental composition of original sample should be provided, such as in [Adv. Mater. 32 (2020) 2000385; Prog. Mater. Sci. 105 (2019) 100576].

Response: We thank the reviewer for this comment. An oxidation reaction occurs at the anode of the sample as it is electrochemically etched. Al, Zr, and Cu successively lost electrons and dissolved, forming pits on the surface of the sample, and then Cu quickly deposited back into the pits. Finally, together with the oxidation product, micro-nano composite structures are formed on the surface of the sample. The chemical composition of the sample surface before and after the reaction was characterized by XRD. Due to the long-range disorder and short-range order of amorphous materials, their XRD patterns significantly differ from crystalline materials. Figure 2 (a) represents the XRD pattern of the sample surface before the reaction, showing that the sample exhibited only one scattering peak. In contrast, Figure 2 (b) shows that the sample surface has characteristic peaks at 43.5°, 50.7° and 74.7°, corresponding to the crystal planes of Cu (111), Cu (200) and Cu (220), respectively. The above data is consistent with the crystallization data of Cu (JCPDS card: 01-1242), indicating that there is copper on the surface of the sample after electrochemical etching.

  1. The quality of the Figures should be further improved, such as the name of X axis in Figure 4b is missing. In addition, there are several typos and mistakes in the present manuscript. Please carefully correct the whole errors. Please do not use an image as the Equation in the manuscript.

Response: We thank the reviewer for this comment. The above problems have been corrected in the manuscript.

Reviewer 2 Report

The present work is interesting and well organized. The use of non-toxic molecules to reach the superhydrophobicity is an important and key factor nowadays. The work seems suitable for publication after the authors address the comments, which are noted in the attached file.

Author Response

To Reviewer 1

Comments to the Author

This is an interesting work that reports a facile electrochemical method for fabrication of stable corrosion-resistant superhydrophobic surface on Zr-based bulk metallic glasses. The corrosion-resistant and superhydrophobic properties were dramatically improved by the electrochemical etching method. The experiments were rationally designed. This work provides a new strategy for improving corrosive and superhydrophobic properties of bulk metallic glasses. It is acceptable in this journal after a minor revision.

  1. The surface morphology evolution at different etching time is characterized by SEM. However, the structural evolution and surface elemental composition are missed out. The XRD pattern of the etched sample and elemental composition of original sample should be provided, such as in [Adv. Mater. 32 (2020) 2000385; Prog. Mater. Sci. 105 (2019) 100576].

Response: We thank the reviewer for this comment. An oxidation reaction occurs at the anode of the sample as it is electrochemically etched. Al, Zr, and Cu successively lost electrons and dissolved, forming pits on the surface of the sample, and then Cu quickly deposited back into the pits. Finally, together with the oxidation product, micro-nano composite structures are formed on the surface of the sample. The chemical composition of the sample surface before and after the reaction was characterized by XRD. Due to the long-range disorder and short-range order of amorphous materials, their XRD patterns significantly differ from crystalline materials. Fig. 3 (a) represents the XRD pattern of the sample surface before the reaction, showing that the sample exhibited only one scattering peak. In contrast, Figure 3 (b) shows that the sample surface has characteristic peaks at 43.5°, 50.7° and 74.7°, corresponding to the crystal planes of Cu (111), Cu (200) and Cu (220), respectively. The above data is consistent with the crystallization data of Cu (JCPDS card: 01-1242), indicating that there is copper on the surface of the sample after electrochemical etching.

  1. The quality of the Figures should be further improved, such as the name of X axis in Figure 4b is missing. In addition, there are several typos and mistakes in the present manuscript. Please carefully correct the whole errors. Please do not use an image as the Equation in the manuscript.

Response: We thank the reviewer for this comment. The above problems have been corrected in the manuscript.

To Reviewer 2

Comments to the Author

The present work is interesting and well organized. The use of non-toxic molecules to reach the superhydrophobicity is an important and key factor nowadays. The work seems suitable for publication after the authors address the comments, which are noted in the attached file.

  1. Page 2, lines 39-40 – “…low-surface-energy materials.” This statement is not completely true in fact, SHS can be produced also without a previous roughening of the surface but using particles already functionalised by spray coating [Ferrari M., Coatings 2019, 9, 303; doi:10.3390/coatings9050303], by spin coating [Tuvshindorj U, ACS Appl Mater Interfaces 2014;6:9680–8. https://doi.org/10.1021/am502117a] or furthermore creating a rough surface from a low surface energy material without a subsequent functionalization [Y.L. Zhan, Colloids Surfaces A Physicochem. Eng. Asp. 2017, 535, 8–15. doi:10.1016/j.colsurfa.2017.09.018.].Only some ref. as examples were reported.

Response: We thank the reviewer for this comment. It is indeed possible to achieve superhydrophobicity only through the surface microstructures. Normally, micro-nano-scale structures were first developed on a solid surface. Subsequently, the rough surface was modified using low-surface-energy materials.

  1. Page 3, line 95 – “stearic acid” Add solution concentration and immersion time.

Response: We thank the reviewer for this comment. Finally, they were modified by immersion in 8 mmol/L stearic acid at room temperature for 20 h, followed by heating at 120 ℃ for 2 h.

  1. Page 3, line 109 – Add here the description of the chemical stability test (pH and heating) and about self-cleaning. A description is present only in the results and discussion section.

Response: We thank the reviewer for this comment. In order to evaluate the chemical stability of the sample surface, HCl and NaOH were used to prepare solutions with different pH, and their droplets were dropped onto the surface of the superhydrophobic sample. After standing for 5 minutes, the contact angle between the droplets and the superhydrophobic surface was measured. To further evaluate the durability of the superhydrophobic surface, the samples were placed in oven at different temperatures for 2 h, and the change in CA was measured.

    To prove the self-cleaning ability of the superhydrophobic surface, quartz sand was evenly covered on the sample surface, and water droplets dripped from the top of the inclined surface. In addition, the self-cleaning performance of the superhydrophobic surface was further confirmed by the pulling experiment. This time, sand was placed on the sample surface and 5 μL of water drop was suspended on the probe which was then moved down to wrap the particles in the water droplets and lifted up afterwards.

  1. Page 6, Figure 2 – The letters on the images are not easily readable, especially in B/W. I suggest to set a white background.

Response: We thank the reviewer for this comment. The picture has been modified in the manuscript.

  1. Page 10, line 300 – “contaminants” NaCl cannot be properly defined as a contaminant since it is soluble in water. To better explain this behaviour could be employ dust such as carbon black or other insoluble particles.

Response: We thank the reviewer for this comment. This work has replaced NaCl with quartz sand.

To Reviewer 3

Comments to the Author

The manuscript presents an electrochemical method of preparation of superhydrophobic surfaces based on metallic glasses. The authors provide a characteristic of these surfaces taking under consideration corrosive properties, and superhydrophobicity as well. Due to the possibility of using the tested surfaces in many fields, especially since they are biocompatible, I consider this manuscript to be very interesting and worth publishing.

Some revision suggestions:

  1. The Authors use droplets with a volume of 5 ml. Were other volumes also checked?

Response: We thank the reviewer for this comment. We have also measured the contact angle with 2 μL of water droplets on the surface and got the same result as before. And this volume is also used in other work, such as Yao, Lu, Jinlong, et al. Preparation of Superoleophobic and Superhydrophobic Titanium Surfaces via an Environmentally Friendly Electrochemical Etching Method[J]. ACS Sustainable Chemistry & Engineering, 2012, 1(1):102-109; Song J, Xu W, Lu Y, et al. Rapid fabrication of superhydrophobic surfaces on copper substrates by electrochemical machining[J]. Applied Surface Science, 2011, 257(24):10910-10916; Gao Y, Sun Y, Guo D. Facile fabrication of superhydrophobic surfaces with low roughness on Ti–6Al–4V substrates via anodization[J]. Applied Surface Science, 2014, 314:754-759.

  1. Page 7, lines 193-201 – The chemical reactions are not stoichiometric. Please correct it.

Response: We thank the reviewer for this comment.

Al(s) + 3Cl- (aq) = AlCl3 (aq) + 3e-

Zr(s) + 4Cl- (aq) = ZrCl4 (aq) + 4e-

Cu(s) + Cl- (aq) = CuCl (s) + e-

2CuCl (s) + H2O (l) = Cu2O (s) + 2Cl- (aq) + 2H+ (aq)

2Cl- + 2e- = Cl2

Cu2O (s) + 2H2O (l) + O2 = 2CuO (s) + 2H2O2 (l)

Cu2O (s) + 2H2O2 (l) = 2CuO (s) + 2H2O (l)

2H+ + 2e- = H2

  1. Page 6, Figure 2 – I would add the values of CA near the droplet – it would make the figure more self-explanatory.

Response: We thank the reviewer for this comment. The picture has been modified in the manuscript.

  1. Page 8, line 227 – the equation seems to be in a very low resolution. Please re-write it.

Response: We thank the reviewer for this comment. This problem has been corrected in the manuscript.

  1. Page 8, line 250 – could the Authors provide the equivalent circuit used?

Response: We thank the reviewer for this comment. The data fitted by ZView2 software is shown in Table 3, and the equivalent circuit used is provided in Figure 4 (d).

  1. Page 9, Figure 4 b – the x axis does not have a description – seems like it’s been cut off. Please correct it.

Response: We thank the reviewer for this comment. The picture has been modified in the manuscript

  1. Page 10, line 280 – “…the CA was measured.” What exactly do the Authors mean? Water contact angle or the solutions of different pH? Maybe the sentence should be re-written to avoid misunderstanding.

Response: We thank the reviewer for this comment. In order to evaluate the stability of the sample surface, HCl and NaOH were used to prepare solutions with different pH, and their droplets were dropped onto the surface of the superhydrophobic sample. After standing for 5 min, the contact angle between the droplets and the superhydrophobic surface was measured.

  1. Page 10, line 291 – Shouldn't the Self-cleaning propertiestitle start with a capital letter?

Response: We thank the reviewer for this comment. This problem has been corrected in the manuscript.

  1. Page 10, line 301 – The Authors refer to “low adhesion” - but was this phenomenon really studied? Have the adhesion parameters been determined? I think that since the contact angle measurements were made, it could be done with little effort, which would undoubtedly enrich the work.

Response: We thank the reviewer for this comment. For superhydrophobic surfaces, not only a large contact angle is required, but also the adhesion of the droplets on the solid surface is required to be small, that is, the droplets are easy to roll off the surface. Some factors such as adhesion and chemical unevenness in the process of droplets contacting the solid surface can cause adhesion hysteresis. The sliding angle can more intuitively reflect the difficulty of the droplet rolling on the solid surface. The smaller the sliding angle, the easier it is for the droplet to leave the surface, reflecting the lower surface adhesion. The sliding angle in this work can be found in Table 1.

  1. Page 11, Figure 6 b – the inscription "NaCl partical" has a typo on one side and is poorly legible on the other. Nevertheless, I understood from the content of the article that sand particles were placed on the slides, which were then cleaned by drops. So where do these NaCl particles come from?

Response: We thank the reviewer for this comment. This work has replaced NaCl with quartz sand. Put the quartz sand on the surface of the sample, then 5 μL of water drop was suspended on the probe, moving the probe downward so that the particles were wrapped in the water droplets. The probe was lifted and the particles were also carried away from the sample surface.

  1. In section "References” – positions: 1- 10 are missing page numbers and article titles.

Response: We thank the reviewer for this comment. This problem has been corrected in the manuscript.

To Reviewer 4

Comments to the Author

Recommendation:  this paper is worth publication after properly addressing the following issues.

  1. Please explain further how the Cassie-baxter model supports their results.

 Response: We thank the reviewer for this comment. A complex contact idea proposed by Cassie and Baxter assumes that when a water droplet is in contact with a solid surface, it forms a composite contact, combining solid–liquid contact with gas–liquid contact. The expression is as follows:

  cosθc =fs (cosθ + 1) -1

Where fs is the area fraction occupied by the solid-liquid contact surface. It can be analyzed from the formula that the smaller the fs, the larger the contact angle. Accordingly, under the combined effect arising from the presence of micro-nano composite structures and the surface tension of water droplets, cavitation occurs on the rough surface. The trapped air leads to a decrease in fs, which improves the wettability and increases the contact angle.

  1. After being immersed in stearic acid for 20 hours, what chemical change on the surface level occurred? Why could the immersion in stearic acid lower surface energy?

Response: We thank the reviewer for this comment. After chemical modification, self-assembled monolayers (SAMs) of stearic acid are formed on the surface of the sample. The hydrophilic polar head-group (-COOH) of stearic acid is tightly bonded to the substrate through chemisorption, leading to the hydrophobic end-group (-CH3) staying away from the surface and decreased surface energy. The reaction processes may be as follows:

Al3+ + 3CH3(CH2)16COOH = Al[CH3(CH2)16COO] + 3H+

Zr4+ + 4CH3(CH2)16COOH = Zr[CH3(CH2)16COO] + 4H+

Cu2+ + 2CH3(CH2)16COOH = Cu[CH3(CH2)16COO] + 2H+

The related discussion has been added in the revised manuscript. 

  1. Reinterpret what is the temperature-resistant self-cleaning properties.

 Response: We thank the reviewer for this comment. The superhydrophobic surface has thermal stability given that it remains superhydrophobic after being placed in a 210℃ oven for 2 h.

The superhydrophobic surface is proven to be self-cleaning if it allows droplets to take away surface contaminants while rolling down.

  1. More explanations are needed on the case with etching time of 15 min, which yields the highest contact angle.

Response: We thank the reviewer for this comment. At the etching time of 15 min, although the surface morphology reflected the presence of coral-reef-shaped micro-nano composite structures (Fig. 1e), the number of nanoscale mastoids increased significantly (Fig. 1f). At this time, the roughness of the sample surface is 4.566 μm, which is almost twice as large as that at 7 min. According to the Cassie-Baxter expression, when the surface roughness increases, the more air is trapped on the sample surface, so the fs decreases and the contact angle increases. In addition, the dense coral reef-like structure provides a uniform air cushion on the sample surface, resulting in a smaller solid-liquid contact area on the sample surface. Hence the contact angle reaches 154.3° after modification.

Reviewer 3 Report

Manuscript number: molecules-1116757

The manuscript: Facile electrochemical method for fabrication of stable corrosion-resistant superhydrophobic surface on Zr-based bulk metallic glasses

by:

Mengmeneg Yu, Ming Zhang, Jing Sun, Feng Liu, Yujia Wang, Guanzhong Ding, Xiubo Xie, Li Liu, Xiangjin Zhao, Haihong Li

The manuscript presents an electrochemical method of preparation of superhydrophobic surfaces based on metallic glasses . The authors provide a characteristic of these surfaces taking under consideration corrosive properties, and superhydrophobicity as well. Due to the possibility of using the tested surfaces in many fields, especially since they are biocompatible, I consider this manuscript to be very interesting and worth publishing.

Some revision suggestions:

  1. The Authors use droplets with a volume of 5 ml. Were other volumes also checked?
  2. Page 7, lines 193-201 – The chemical reactions are not stoichiometric. Please correct it.
  3. Page 6, Figure 2 – I would add the values of CA near the droplet – it would make the figure more self-explanatory.
  4. Page 8, line 227 – the equation seems to be in a very low resolution. Please re-write it.
  5. Page 8, line 250 – could the Authors provide the equivalent circuit used?
  6. Page 9, Figure 4 b – the x axis does not have a description – seems like it’s been cut off. Please correct it.
  7. Page 10, line 280 – “…the CA was measured.” What exactly do the Authors mean? Water contact angle or the solutions of different pH? Maybe the sentence should be re-written to avoid misunderstanding.
  8. Page 10, line 291 – Shouldn't the Self-cleaning properties title start with a capital letter?
  9. Page 10, line 301 – The Authors refer to “low adhesion” - but was this phenomenon really studied? Have the adhesion parameters been determined? I think that since the contact angle measurements were made, it could be done with little effort, which would undoubtedly enrich the work.
  10. Page 11, Figure 6 b – the inscription "NaCl partical" has a typo on one side and is poorly legible on the other. Nevertheless, I understood from the content of the article that sand particles were placed on the slides, which were then cleaned by drops. So where do these NaCl particles come from?
  11. In section "References” – positions: 1- 10 are missing page numbers and article titles.

Author Response

  1. The Authors use droplets with a volume of 5 ml. Were other volumes also checked?

Response: We thank the reviewer for this comment. We have also measured the contact angle with 2 μL of water droplets on the surface and got the same result as before. And this volume is also used in other work, such as Yao, Lu, Jinlong, et al. Preparation of Superoleophobic and Superhydrophobic Titanium Surfaces via an Environmentally Friendly Electrochemical Etching Method[J]. ACS Sustainable Chemistry & Engineering, 2012, 1(1):102-109; Song J, Xu W, Lu Y, et al. Rapid fabrication of superhydrophobic surfaces on copper substrates by electrochemical machining[J]. Applied Surface Science, 2011, 257(24):10910-10916; Gao Y, Sun Y, Guo D. Facile fabrication of superhydrophobic surfaces with low roughness on Ti–6Al–4V substrates via anodization[J]. Applied Surface Science, 2014, 314:754-759.

  1. Page 7, lines 193-201 – The chemical reactions are not stoichiometric. Please correct it.

Response: We thank the reviewer for this comment.

Al(s) + 3Cl- (aq) = AlCl3 (aq) + 3e-

Zr(s) + 4Cl- (aq) = ZrCl4 (aq) + 4e-

Cu(s) + Cl- (aq) = CuCl (s) + e-

2CuCl (s) + H2O (l) = Cu2O (s) + 2Cl- (aq) + 2H+ (aq)

2Cl- + 2e- = Cl2

Cu2O (s) + 2H2O (l) + O2 = 2CuO (s) + 2H2O2 (l)

Cu2O (s) + 2H2O2 (l) = 2CuO (s) + 2H2O (l)

2H+ + 2e- = H2

  1. Page 6, Figure 2 – I would add the values of CA near the droplet – it would make the figure more self-explanatory.

Response: We thank the reviewer for this comment. The picture has been modified in the manuscript.

  1. Page 8, line 227 – the equation seems to be in a very low resolution. Please re-write it.

Response: We thank the reviewer for this comment. This problem has been corrected in the manuscript.

  1. Page 8, line 250 – could the Authors provide the equivalent circuit used?

Response: We thank the reviewer for this comment. The data fitted by ZView2 software is shown in Table 3, and the equivalent circuit used is provided in Figure 4 (d).

  1. Page 9, Figure 4 b – the x axis does not have a description – seems like it’s been cut off. Please correct it.

Response: We thank the reviewer for this comment. The picture has been modified in the manuscript

  1. Page 10, line 280 – “…the CA was measured.” What exactly do the Authors mean? Water contact angle or the solutions of different pH? Maybe the sentence should be re-written to avoid misunderstanding.

Response: We thank the reviewer for this comment. In order to evaluate the stability of the sample surface, HCl and NaOH were used to prepare solutions with different pH, and their droplets were dropped onto the surface of the superhydrophobic sample. After standing for 5 min, the contact angle between the droplets and the superhydrophobic surface was measured.

  1. Page 10, line 291 – Shouldn't the Self-cleaning propertiestitle start with a capital letter?

Response: We thank the reviewer for this comment. This problem has been corrected in the manuscript.

  1. Page 10, line 301 – The Authors refer to “low adhesion” - but was this phenomenon really studied? Have the adhesion parameters been determined? I think that since the contact angle measurements were made, it could be done with little effort, which would undoubtedly enrich the work.

Response: We thank the reviewer for this comment. For superhydrophobic surfaces, not only a large contact angle is required, but also the adhesion of the droplets on the solid surface is required to be small, that is, the droplets are easy to roll off the surface. Some factors such as adhesion and chemical unevenness in the process of droplets contacting the solid surface can cause adhesion hysteresis. The sliding angle can more intuitively reflect the difficulty of the droplet rolling on the solid surface. The smaller the sliding angle, the easier it is for the droplet to leave the surface, reflecting the lower surface adhesion. The sliding angle in this work can be found in Table 1.

  1. Page 11, Figure 6 b – the inscription "NaCl partical" has a typo on one side and is poorly legible on the other. Nevertheless, I understood from the content of the article that sand particles were placed on the slides, which were then cleaned by drops. So where do these NaCl particles come from?

Response: We thank the reviewer for this comment. This work has replaced NaCl with quartz sand. Put the quartz sand on the surface of the sample, then 5 μL of water drop was suspended on the probe, moving the probe downward so that the particles were wrapped in the water droplets. The probe was lifted and the particles were also carried away from the sample surface.

  1. In section "References” – positions: 1- 10 are missing page numbers and article titles.

Response: We thank the reviewer for this comment. This problem has been corrected in the manuscript.

Reviewer 4 Report

Title:  Facile electrochemical method for fabrication of stable corrosion-resistant superhydrophobic surface on Zr-Based bulk metallic glasses

 By M.Yu, et al

Recommendation:   this paper is worth publication after properly addressing the following issues.

  1. Please explain further how the Cassie-baxter model supports their results.

  1. After being immersed in stearic acid for 20 hours, what chemical change on the surface level occurred? Why could the immersion in stearic acid lower surface energy?

  1. Reinterpret what is the temperature-resistant self-cleaning properties.

  1. More explanations are needed on the case with etching time of 15 min, which yields the highest contact angle.

Author Response

  1. Please explain further how the Cassie-baxter model supports their results.

 Response: We thank the reviewer for this comment. A complex contact idea proposed by Cassie and Baxter assumes that when a water droplet is in contact with a solid surface, it forms a composite contact, combining solid–liquid contact with gas–liquid contact. The expression is as follows:

cosθ=fs (cosθ + 1) -1  

Where fs is the area fraction occupied by the solid-liquid contact surface. It can be analyzed from the formula that the smaller the fs, the larger the contact angle. Accordingly, under the combined effect arising from the presence of micro-nano composite structures and the surface tension of water droplets, cavitation occurs on the rough surface. The trapped air leads to a decrease in fs, which improves the wettability and increases the contact angle.

  1. After being immersed in stearic acid for 20 hours, what chemical change on the surface level occurred? Why could the immersion in stearic acid lower surface energy?

Response: We thank the reviewer for this comment. After chemical modification, self-assembled monolayers (SAMs) of stearic acid are formed on the surface of the sample. The hydrophilic polar head-group (-COOH) of stearic acid is tightly bonded to the substrate through chemisorption, leading to the hydrophobic end-group (-CH3) staying away from the surface and decreased surface energy. The reaction processes may be as follows:

Al3+ + 3CH3(CH2)16COOH = Al[CH3(CH2)16COO] + 3H+

Zr4+ + 4CH3(CH2)16COOH = Zr[CH3(CH2)16COO] + 4H+

Cu2+ + 2CH3(CH2)16COOH = Cu[CH3(CH2)16COO] + 2H+

The related discussion has been added in the revised manuscript. 

  1. Reinterpret what is the temperature-resistant self-cleaning properties.

 Response: We thank the reviewer for this comment. The superhydrophobic surface has thermal stability given that it remains superhydrophobic after being placed in a 210℃ oven for 2 h.

The superhydrophobic surface is proven to be self-cleaning if it allows droplets to take away surface contaminants while rolling down.

  1. More explanations are needed on the case with etching time of 15 min, which yields the highest contact angle.

Response: We thank the reviewer for this comment. At the etching time of 15 min, although the surface morphology reflected the presence of coral-reef-shaped micro-nano composite structures (Fig. 1e), the number of nanoscale mastoids increased significantly (Fig. 1f). At this time, the roughness of the sample surface is 4.566 μm, which is almost twice as large as that at 7 min. According to the Cassie-Baxter expression, when the surface roughness increases, the more air is trapped on the sample surface, so the fs decreases and the contact angle increases. In addition, the dense coral reef-like structure provides a uniform air cushion on the sample surface, resulting in a smaller solid-liquid contact area on the sample surface. Hence the contact angle reaches 154.3° after modification.
